# The Influence of Replacing Aggregates and Cement by LFS on the Corrosion of Steel Reinforcements

**María Isabel Prieto** [1],*, **María de las Nieves González** [2],*, **Ángel Rodríguez** [3] **and Alfonso Cobo** [1]

1   Tecnología de la Edificación, Universidad Politécnica de Madrid, 28040 Madrid, Spain; alfonso.cobo@upm.es
2   Construcciones Arquitectónicas y su Control, Universidad Politécnica de Madrid, 28040 Madrid, Spain
3   Construcciones Arquitectónicas e Ingeniería de la Construcción y del Terreno, Universidad de Burgos, 09001 Burgos, Spain; arsaizmc@ubu.es
*   Correspondence: mariaisabel.prieto@upm.es (M.I.P.); mariadelasnieves.gonzalez@upm.es (M.d.l.N.G.)

**Abstract:** The aim of this study is to investigate the corrosion behavior of steel reinforcement embedded in mortar specimens in which both the aggregate and cement are partially replaced by ladle furnace slag (LFS) and different percentages of chloride ions by weight of cement are introduced into the mix at the time of kneading. The corrosion behavior was studied by using electrochemical techniques in order to evaluate the corrosion rate and the symptoms produced in steels of specimens with and without slag LFS. From the analysis of the results, it is concluded that the use of LFS in a partial replacement of aggregate and cement in mortar specimens does not compromise the behavior of the mortar with regard to corrosion of the steel reinforcement; consequently, partial replacement by LFS is fully feasible from this standpoint.

**Keywords:** corrosion; reinforcements; concrete; slag; LFS

## 1. Introduction

Due to its versatility and low cost, reinforced concrete is the most frequently used structural material in the field of building and engineering; it reaches annual volumes of 10 km$^3$/year and causes major environmental problems. The manufacture of Portland cement produces between 5% and 8% of global $CO_2$ emissions due to human activity [1–3].

The regulations [4–7] specify the importance of durability in reinforced concrete and take as a determining factor the corrosion of embedded reinforcement steel.

Corrosion of embedded steel reinforcement may occur through carbonation of the concrete cover, the presence of chlorides, or the combination of both. These are the factors which trigger the corrosion process and transfer the reinforcement from the passive state, in which the corrosion rates are barely significant, to the active state, in which corrosion affects the durability of concrete structures.

Besides these triggering factors, the continuous presence of oxygen and humidity is necessary as these are the controlling factors for the rate of the corrosion process. In practice, the presence of chlorides in concrete is a determining factor in the onset of reinforcement corrosion. In general, a threshold level of 0.4% of chloride ions by weight of cement is considered, as proposed by RILEM Committee 60-CSC [8].

Moreover, the particular feature of steel as an indefinitely recyclable material has meant that the use of scrap iron has become increasingly widespread in steel production at a global level. World crude steel production reached 1689.4 million tons (Mt) for the year 2017 [9]. Steel production in Spain has been increasing over the years, reaching 14.5 million tons in 2017 [10]. This provides major environmental benefits as a ton of steel manufactured in electric arc furnaces (EAF) consumes

only 9–12 GJ/tcs, with a consequent reduction in $CO_2$ emissions, though the continuous increase in steel production has been the cause of a significant increase in environmental problems over the years [11,12].

EAF steel production generates approximately 120 to 180 kg of black slag and 20 to 30 kg of white slag for each ton of steel produced [13]. The volume of ladle furnace slag (LFS) generated in the European Union was 2.0 million tons in 2016 [14], while in Spain, with 70.5% of the steel produced by EAF, the volume of white slags was between 0.20 and 0.31 million tons in 2017 [10].

Among the most immediate applications of the LFS are the manufacture of clinker, the partial substitution of aggregates and/or cement in mortars and concretes, and the stabilization of soils and pavements [15,16]. The characteristics of the slag that can be used in concrete depend on the different treatments they have received [17].

There is some research on the characterization of ladle furnace slags and their hydration properties. These studies show that the LFSs are a dusty material and are composed mainly of calcium oxide (50%), silicon (15%), aluminum (12%), and magnesium (9%). Its density is of the order of 2.7 g/cm$^3$ and associated important volumetric expansions are due to the high proportion of free lime. If this expansion is controlled, it does not imply any inconvenience for its use in construction materials [18–20]. There has also been some research that studied the behavior of mortars or concretes in which the aggregate and/or the cement was partially replaced by LFS and that indicate that its application is viable, improving the properties of mortars and sustainability both in traditional concretes and in self-compacting concretes [21–27].

Taking into account the previous premises, it is observed that although the corrosion process of steel bars embedded in concrete has been studied in various research projects [28–30], no studies have analyzed this process when ladle furnace slag is introduced into the concrete instead of some of its components. In this work, corrosion behavior was tested using electrochemical techniques in order to evaluate the corrosion rate and the effects produced in the reinforcements of the specimens with and without LFS. These specimens were initially subjected to a natural corrosion process and then to accelerated corrosion; this allowed us to determine the influence range of the LFS slag.

## 2. Experimental Investigation

### 2.1. Materials

The materials employed in the manufacturing of the mortar specimens were Portland cement CEM I/42.5 R [31] (Table 1), silica sand specified as "siliceous rolled aggregate 0/4" (Table 2), urban drinking water, retardant SikaTard, calcium chloride ($CaCl_2$), and ladle furnace slag (Table 3).

**Table 1.** Main properties of Portland cement.

| Chemical (%) | | Physical | |
|---|---|---|---|
| $SO_3$ | 3.4 | Specific surface area (Blaine) | 414 m$^2$/kg |
| $Cl^-$ | 0.01 | Density | 3.15 g/cm$^3$ |
| Calcination loss | 1.72 | Le Chatelier expansion | 0 mm |
| Insoluble residue | 0.4 | Initial setting time | 108 min |
| | | Final setting time | 160 min |

**Table 2.** Characterization of the sand used in the specimens.

| | |
|---|---|
| Aggregate | 0.78% |
| Sand equivalent | 78 |
| Real density | 2.619 g/cm$^3$ |
| Normal absorption coefficient | 15% |
| Saturated surface dry density | 2.630 g/cm$^3$ |
| Clay lumps | 0.01% |
| Low specific weight particles | 0.00% |
| Coefficient of type of course | 0.26% |
| Soft particles | 0.93% |
| S, S03, Cl$^-$ | 0% |

**Table 3.** Physical properties and chemical composition of ladle furnace slag (LFS).

| | |
|---|---|
| Density | 2.83 g/cm$^3$ |
| Specific surface | 2064 cm$^2$/g |
| Chlorides | Absence |
| Total sulfur, expressed as sulfate ions | <1% |
| Clay lumps | Absence |
| Organic material | Absence |
| CaO | 56% |
| SiO$_2$ | 17% |
| Al$_2$O$_3$ | 11% |
| MgO | 10% |
| Others (Fe$_2$O$_3$+MnO+TiO$_2$+SO$_3$+Na$_2$O+K$_2$O) | 6% |

Taking into account the results obtained in other research work [25], the following guidelines for dosages were considered:

- A ratio of cement/sand/water by weight of 1:6:$w$, with $w$ relating to the necessary quantity of water to achieve a slump of 175 ± 10 mm;

- A compressive mechanical strength at 28 days of at least 7.5 N/mm$^2$;

- 30% of cement and 25% of sand substituted for ladle furnace slag in the LFS specimens;

- 0.5% by weight of cement of retardant was used.

Two series of mortar test specimens were manufactured: one used as a standard test specimen (MCC) and another in which part of the cement and sand was replaced by LFS (MCE) (Table 4). During kneading, chloride ions were added in the form of calcium chloride, corresponding to 0%, 0.4%, 0.8%, 1.2%, and 2% by weight of cement.

**Table 4.** Dosages of the samples with and without LFS.

| Label | Cement (g) | Sand (g) | Water (g) | LFS (g) | Retardant (g) | CaCl$_2$ (g) | Ion Cl$^-$ (%) |
|---|---|---|---|---|---|---|---|
| MCC-0.0 | 99.6 | 597.5 | 102.9 | —— | 0.50 | 0.00 | 0.0 |
| MCC-0.4 | 99.6 | 597.5 | 102.9 | —— | 0.50 | 0.80 | 0.4 |
| MCC-0.8 | 99.6 | 597.5 | 102.9 | —— | 0.50 | 1.60 | 0.8 |
| MCC-1.2 | 99.6 | 597.5 | 102.9 | —— | 0.50 | 2.40 | 1.2 |
| MCC-2.0 | 99.6 | 597.5 | 102.9 | —— | 0.50 | 3.99 | 2.0 |
| MCE-0.0 | 68.6 | 441.1 | 113.9 | 176.4 | 0.35 | 0.00 | 0.0 |
| MCE-0.4 | 68.6 | 441.1 | 113.9 | 176.4 | 0.35 | 0.55 | 0.4 |
| MCE-0.8 | 68.6 | 441.1 | 113.9 | 176.4 | 0.35 | 1.10 | 0.8 |
| MCE-1.2 | 68.6 | 441.1 | 113.9 | 176.4 | 0.35 | 1.65 | 1.2 |
| MCE-2.0 | 68.6 | 441.1 | 113.9 | 176.4 | 0.35 | 2.75 | 2.0 |

## *2.2. Experimental Setup*

This experimental work involved the preparation of prismatic mortar test specimens measuring $6 \times 8 \times 2$ cm$^3$, in which 3 parallel steel rebars with a diameter of 6 mm were embedded (Figure 1). The study is a continuation of research into LFS as a partial replacement for aggregate and cement in mortars [32,33] from the viewpoint of its behavior with regard to rebar corrosion and the symptom this produces with the time.

All the steel bars were measured, numbered, and weighed before their introduction into the mold. The steel–concrete–atmosphere interface was protected by adhesive tape over a length of 5 cm in order to avoid the possibility of localized corrosion attacks by differential aeration, leaving an effective length of 6 cm contact between the steel and the concrete. The mortar was kneaded in a mechanical mixer for 90 s. The mold was filled in two pourings, coinciding with the arrangement of the steel bars. Thereafter, test specimens were placed in a humidity chamber for 24 h. After this period, the specimens were demolded and placed again in the humidity chamber for 28 days, and then they were left to dry normally. With the aim of studying the corrosion of the bars by using electrochemical techniques, the specimens were once again placed in a humidity chamber for 550 days, at a temperature of 20 °C and a relative humidity 95%, associating each measurement of corrosion potential and corrosion rate with the degree of moistness of the mortar.

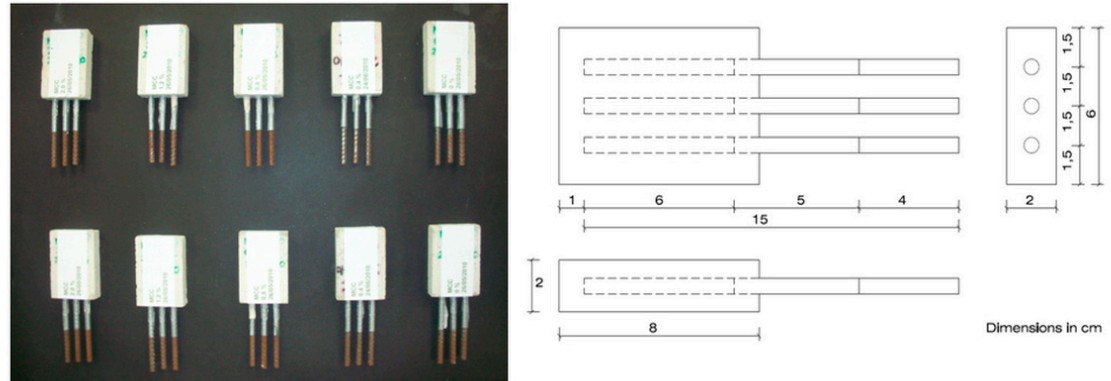

**Figure 1.** Geometrical characteristics of the specimens and specimens corresponding to the MCC dosages.

To highlight the symptoms that rebar corrosion produces in the mortar test specimens with mixed-in chlorides, the corrosion process was further accelerated by connecting the central bar of the standard test specimens as well as the test specimens containing LFS, both having mixed-in chloride ions percentages of 1.2% and 2%, to a direct current source. Once connected, a constant anodic current of 1.3 mA, corresponding to 10 µA/cm$^2$, was passed through each central bar for 282 days. During the periodic measurements of the voltage needed to maintain this preset current and observation of the symptoms, the degree of moistness of the test specimens, with measurements taken before and after their moistening on each occasion, was again associated with the stabilized potential.

## *2.3. Measurement Techniques*

Electrochemical measurements were performed with an AUTOLAB/PGSTAT302N potentiostat (Version 4.9 AUTO83745, ECO CHEMIE, Utrech, Netherland, 2008) [34]. The central steel bar was used as the working electrode while the parallel bars on the sides were used as the counter electrodes. For the electrochemical measurements a silver/silver chloride reference electrode was used (SSCE; +0.222 V SHE). During the measurements a damp flannel was placed on the mortar surface in order to improve electrolytic contact between the mortar and reference electrode (Figure 2).

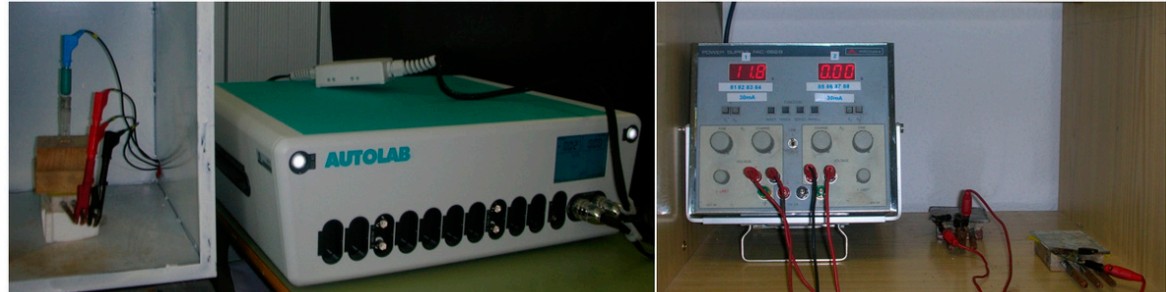

**Figure 2.** Setup for carrying out electrochemical measurements using an AUTOLAB/PGSTAT302N potentiostat and setup of the accelerated corrosion process using a POWER SUPPLY FAC-662B POWER SUPPLY FAC-662B (PROMAX, Barcelona, Spain, 1998).

All the electrochemical measurements were associated with the humidity content of the test specimens, which were gradually moistening and were kept in a humidity chamber.

The use of the potentiostat enabled the evolution of the corrosion to be studied by means of the polarization curve, producing data on the open circuit corrosion potential ($E_{corr}$), and through the polarization resistance on the corrosion rate ($i_{corr}$). Since midway through the 20th century the polarization resistance technique has become a common technique to study corrosion [35].

The first evaluation of rebar behavior with regard to corrosion was made by measuring their corrosion potentials, which are useful for a qualitative determination of whether the steel bars are in an active or a passive state. The data obtained were interpreted according to the ASTM C 876 standard [36], from which, taking into account the reference electrode used, it was concluded that $E_{corr} < -231$ mV would indicate a 90% probability that corrosion exists in the active state, a potential in the region $-231$ mV $< E_{corr} < -91$ mV would indicate uncertainty, and $E_{corr} > -91$ mV would indicate a probability of 10% that corrosion exists in the active state.

The following step was to interpret the results for the rate or intensity of corrosion as this information enables us to quantitatively interpret the active or passive state of the bars.

According to different research [37], $i_{corr} < 0.1$ $\mu A/cm^2$ indicates a passive state, 0.1 $\mu A/cm^2 <$ $i_{corr} < 0.5$ $\mu A/cm^2$ is equivalent to a low level of corrosion, 0.5 $\mu A/cm^2 < i_{corr} < 1$ $\mu A/cm^2$ to a high level of corrosion, and $i_{corr} > 1$ $\mu A/cm^2$ to a very high level of corrosion.

The central bar was connected to a POWER SUPPLY FAC-662B (PROMAX, Barcelona, Spain, 1998) to accelerate the corrosion process in both the standard test specimens and the test specimens with LFS, for specimens containing 1.2% and 2% of mixed-in chloride ions by weight of cement, making the embedded central steel bar act as an anode. On the upper face of the test specimens was placed a damp flannel with a lead sheet on top; this acted as a continuous counter electrode [38]. The voltage necessary to maintain the intensity of the preset current (1.3 mA) was then measured, moistening the test specimens frequently, as the potential varies substantially with the humidity content (Figure 2).

## 3. Results and Analysis

Figure 3a shows the evolution of the corrosion potential ($E_{corr}$) in standard test specimens while Figure 3b provides the data from test specimens with slag for different chloride ion percentages. The corrosion potentials in the standard test specimens for chloride ion percentages equal to or below 0.4% are clearly less negative than those in the test specimens with chloride ion percentages above the limit set by the standards of different countries, which show potentials suggesting the active state. In the test specimens with LFS, regardless of the chloride ion percentages possessed by the test specimens, the corrosion potentials correspond to the active state. In all test specimens the potential becomes more negative over time and, consequently, with their humidity content, as well as with the chloride ion percentages of the test specimens.

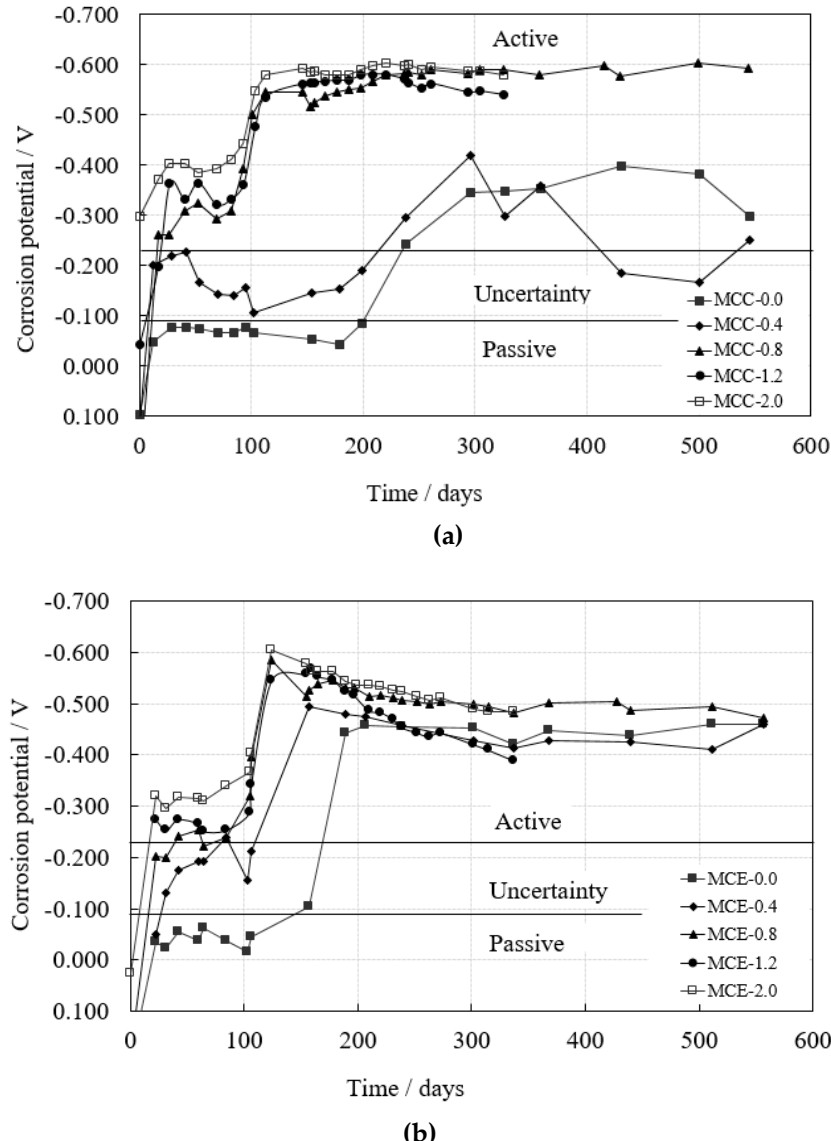

**Figure 3.** Evolution of $E_{corr}$ over time in standard test specimens (**a**) and in LFS specimens (**b**) at different chloride ion percentages.

The corrosion rate increases over time and, therefore, with the humidity content, together with the chloride ion percentage introduced at the time of kneading. In the test specimens with chloride ion percentages within the limit of the EHE "Instrucción de Hormigón Estructural" Instruction (Figure 4a) (0.4% of chloride ions by weight of cement), the corrosion rates are characteristic of the passive state as the maximum values are around 0.1 $\mu A/cm^2$, regardless of whether the test specimens contain LFS, with data corresponding to 545 days of exposure in the humidity chamber. In the test specimens with chloride ion percentages above 0.4% by weight of cement (Figure 4b), the corrosion rates are characteristic of the active state, with values above 1 $\mu A/cm^2$. In the test specimens with 0.8% chloride ions, with data obtained for up to 545 days of exposure, the corrosion rates are slightly higher in the standard test specimens than in the test specimens with LFS. In the test specimens with chloride ion percentages of 1.2% and 2%, data are held on the corrosion rate for up to 326 days of exposure in the moist chamber, as from that moment they underwent an accelerated corrosion process; the results are discussed below. In these test specimens, the corrosion rates increased with the chloride percentage contained in the test specimens, which were slightly higher in the standard test specimens than in the LFS test specimens.

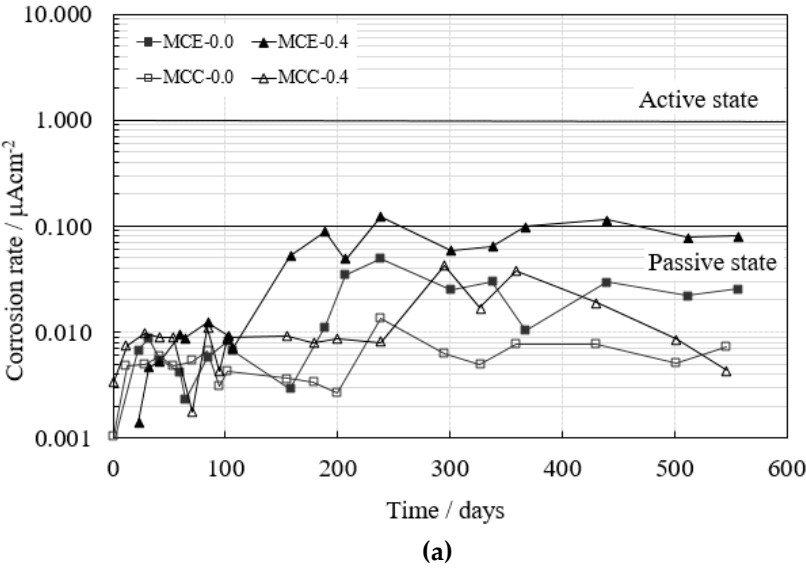

(a)

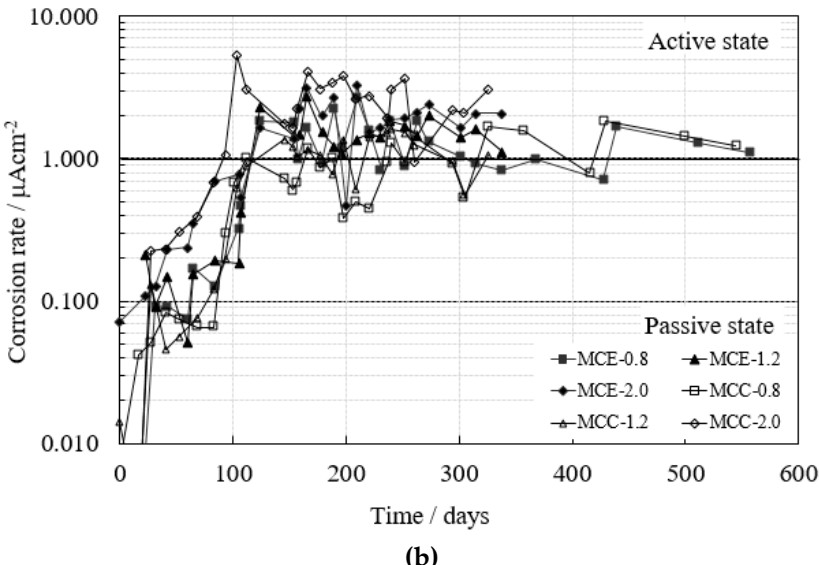

(b)

**Figure 4.** Evolution of $i_{corr}$ over time in test specimens without chlorides and with 0.4% chloride ions by weight of cement (**a**) and in test specimens with chloride ion percentages of 0.8%, 1.2% and 2% (**b**).

Figure 5 shows the evolution of the voltage needed to maintain the preset current at a level of 1.3 mA. The voltage increases over time and with the chloride ion percentage introduced at the time of kneading. Moreover, standard specimens need higher voltage to maintain the preset current than do the specimens with slag LFS.

Figures 6 and 7 show the symptoms produced by the corrosion of the steel bars subjected to an accelerated corrosion process by an impressed anodic current for chloride ion percentages of 1.2% and 2.0% by weight of cement. Each image shows the number of days that had elapsed since the start of the process of natural corrosion, the number of days subjected to accelerated corrosion, and the potential and the intensity of the current passed through the rebar.

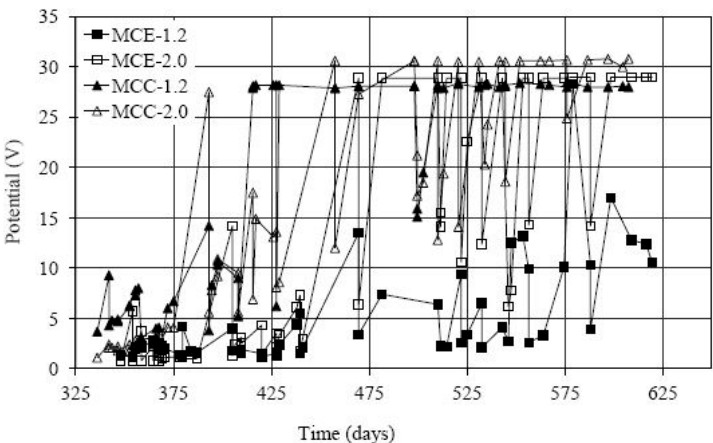

**Figure 5.** Evolution of the potential (V) necessary to maintain a constant current of 1.3 mA in each rebar.

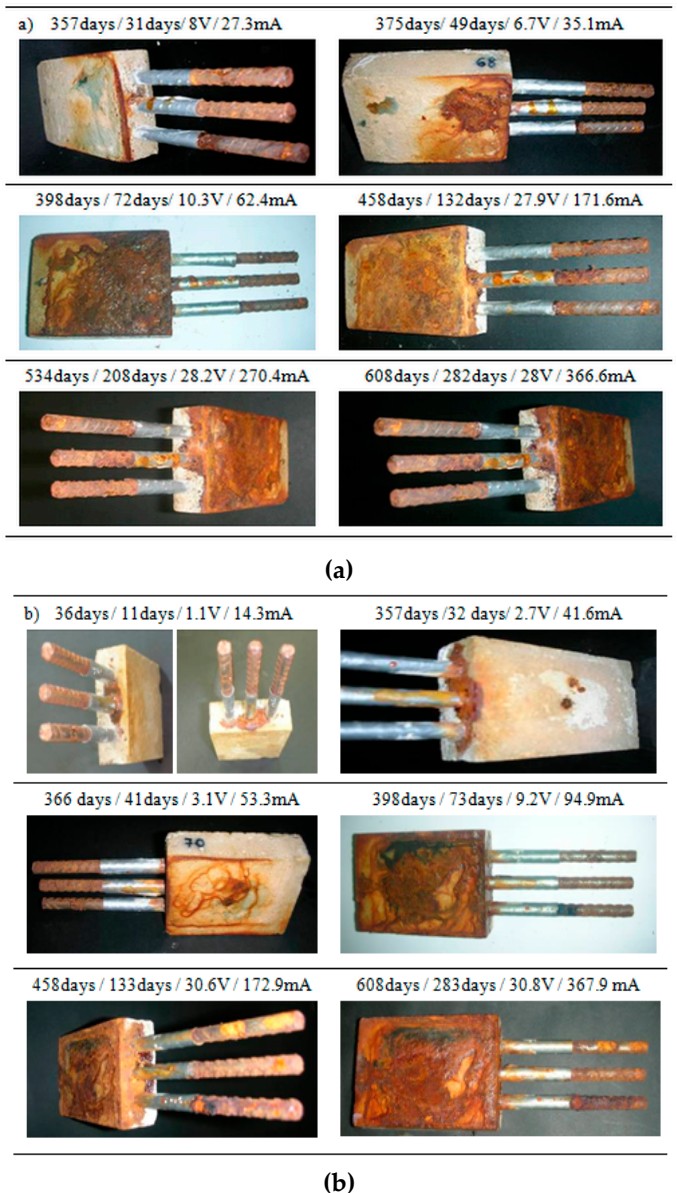

**Figure 6.** Symptoms in the MCC test specimen with 1.2% chloride ions (**a**) and 2% chloride ions (**b**).

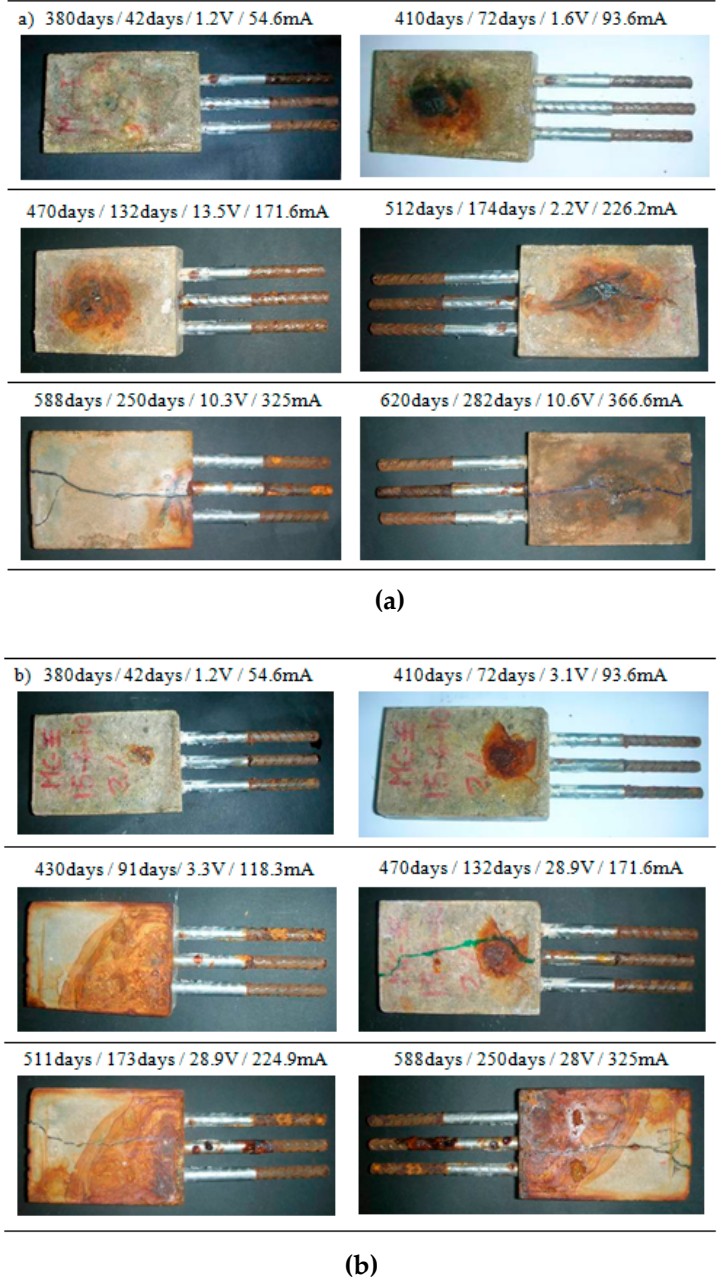

**Figure 7.** Symptoms in the MCE test specimen with 1.2% chloride ions (**a**) and 2% chloride ions (**b**).

In the standard test specimen with a chloride ion percentage of 1.2%, it was observed that the first rust stains began to appear 31 days after the start of the accelerated corrosion, appearing both on the edge of the test specimen and on the lower face, gradually increasing up to 72 days when the stain extended over the entire face of the test specimen, while on the upper face there were no signs of corrosion. The standard test specimen showed no signs of cracking after the central bar had been connected to the current for 282 days, an externally applied electrical charge of 366.6 mA having passed through the central bar (Figure 6a).

In the final test specimen which was subjected to the accelerated corrosion process, corresponding to the other standard test specimen with 2% chloride ions, rust stains began to appear on the edge of the test specimen 11 days after the start of the process, with stains appearing on the lower face of the test specimen after 32 days. From that moment, the rust stains became more uniform on the lower face of the test specimen after 73 days with an electrical charge passed through of 94.9 mA. The test

specimen continued to show signs of corrosion over time in the form of increasingly marked stains, but it presented no cracks after 283 days of accelerated corrosion with an electrical charge of 367.9 mA having passed through the bar (Figure 6b).

In the test specimens with LFS and with 1.2% chloride ions by weight of cement, rust stains began to appear in the upper part of the test specimen coinciding with the position of the bar which was undergoing accelerated corrosion. This initial stain appeared approximately 40 days following the commencement of the accelerated corrosion process and continued to enlarge its surface area and importance with the progressive passage of the current. After 174 days of current, the first cracks appeared, on both the upper and lower faces of the test specimen and even on its edge, coinciding with the central bar. The cracks increased in length and width over time, forking at the end of the test specimen 250 days after the commencement of accelerated corrosion, an overall electrical charge of 325 mA having passed through the bar. The crack width after 282 days, once a charge of 366.6 mA had passed through the bar, was 1 mm (Figure 7a).

The first rust stain in the test specimen with 2% chloride ions and with LFS appeared 42 days after the commencement of accelerated corrosion. The first stain, on the upper face of the test specimen, increased in size over time, leaving an appearance of generalized stains on the test specimen. After 132 days since the commencement of the accelerated corrosion process, the first cracks appeared, coinciding with the central bar and appearing both on the upper and lower faces of the test specimen. The width of the cracks gradually increased; after 250 days and after passing a 325 mA electrical charge through the central bar, numerous cracks in a mesh were formed on the rear of the test specimen, which caused the mortar to disintegrate (Figure 7b).

## 4. Discussion

Both the qualitative results obtained through the measurement of corrosion potential and the quantitative results deduced from the corrosion rates (Figures 3 and 4) show that for chloride ion percentages of 0.4%, the bars are in a passive state but with higher corrosion rates in test specimens with LFS than in the standard test specimens. For test specimens with chloride ion percentages above 0.4%, the bars demonstrate very high states of corrosion both in standard test specimens and in those with LFS, being slightly larger in the former.

The symptoms produced by corrosion show as growing rust stains appearing over time and, therefore, with the increase in current in the standard test specimens, appearing first in the test specimens with 2% chloride ions. The stains began to coincide with the bar through which the external current passed, only to extend gradually along the edge and the lower face of the test specimens, but without cracking 608 days after the start of the experiment and having undergone 283 days of accelerated corrosion.

In the test specimens with LFS, the rust stains began to form later than in the standard test specimens, on the upper face of the test specimen, coinciding with its central bar. In the test specimens with 1.2% chloride ions, 174 days after the commencement of accelerated corrosion, cracks appeared on the lower face and on the edge of the test specimens; these increased in width and length over time until 282 days, when the upper face cracked. In the test specimens with 2% chloride ions by weight of cement, the cracks began 132 days after the commencement of the accelerated corrosion on both the upper and lower faces and on the edge of the test specimens, coinciding with the central bar of the test specimen. The passage of time and the increase in current increased the cracks, with a mesh of cracks appearing in the lower part of the test specimen after 250 days.

Breaking of the test specimens enabled observation of the penetration of corrosive products on the mortar pores in the test specimens with 2% chloride ions; this showed corrosion in the form of pitting produced by the presence of chlorides, beginning with the loss of corrugation in the bar and continuing with the loss of sections in a localized manner (Figure 8).

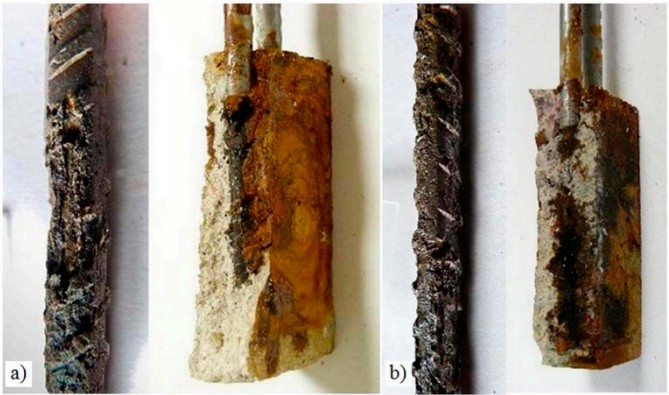

**Figure 8.** Corrosion products in the test specimens without (**a**) and with LFS (**b**), with 2% chloride ions by weight of cement.

Figure 9 shows the corrosion rates in standard test specimens and those with LFS for different chloride ion percentages within 546 days after the experiment onset. The results show that in test specimens with chloride ion percentages of 0.4%, the corrosion rates were slightly higher in the slag test specimens than in the standard ones. By contrast, for chloride ion percentages higher than the EHE Instruction Limit, the corrosion rates were similar and even higher in standard test specimens.

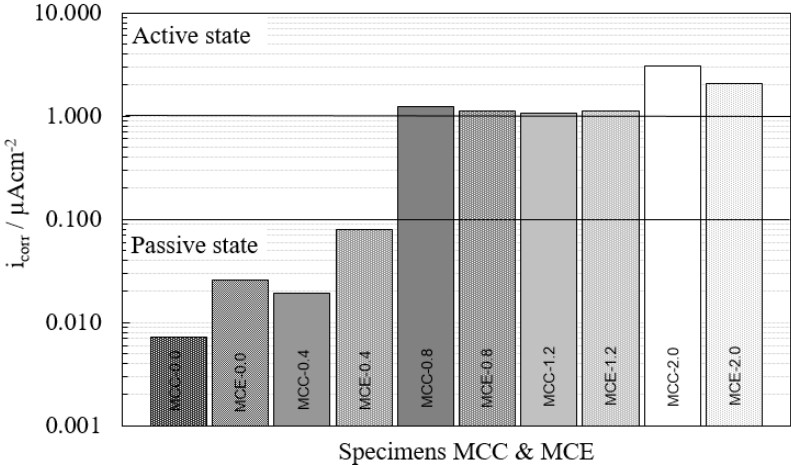

**Figure 9.** $i_{corr}$ values of test specimens attacked by chlorides, in standard test specimens (MCC) and test specimens with LFS (MCE) after 546 days.

## 5. Conclusions

Corrosion potentials both in standard test specimens and in test specimens with LFS became more negative as the humidity of the test specimens and the percentage of chloride ions increased, reaching values which indicate a probability of corrosion which is over 90% for chloride ion percentages above 0.4% by weight of cement.

The corrosion rates increased with the humidity content of the test specimens and with the chloride ion percentage introduced at the time of kneading, both in standard test specimens and in those with LFS, reaching values characteristic of the active state for chloride ion percentages above 0.4% by weight of cement.

The standard test specimens offered greater resistance to the passage of an electrical current than the LFS test specimens. The voltage necessary to maintain a constant current fell drastically on increasing the moisture content, increasing progressively as the degree of moistness of the test specimens decreased.

The symptoms produced in standard test specimens with chloride ion percentages of 1.2% and 2.0% by weight of cement subjected to an accelerated corrosion process consisted of rust stains, initially coinciding with the central bar, which progressively lengthened until they occupied the lower face and the edge of the test specimens.

The symptoms produced in test specimens with LFS with chloride ion percentages of 1.2% and 2.0% by weight of cement that underwent the accelerated corrosion process began with the appearance of rust stains that coincided with the central bar of the test specimens and spread over the upper and lower faces and the edge of the test specimens. The pressure exerted by the corrosive products caused progressive cracking of the test specimens, becoming more evident in the test specimens with a greater chloride ion percentage.

The presence of chlorides in the test specimens produced localized pitting in the steel rebar, both in the standard test specimens and in the test specimens with LFS.

From the study on the corrosion behavior through electrochemical techniques and the study of the symptoms produced in the accelerated corrosion process, it may be concluded that in mortars with the presence of mixed-in chlorides, the incorporation of LFS in the mortar at the time of kneading does not negatively affect the corrosive process of the rebar.

**Author Contributions:** M.I.P.: experimental tests, formal analysis, investigation, and writing—original draft preparation; A.C.: conceptualization, methodology, project administration, supervision, and discussion of the results; A.R.: resources, validation, and writing—review; M.d.l.N.G.: methodology, investigation, and writing—review and editing.

**Funding:** This research received no external funding.

**Conflicts of Interest:** The authors declare no conflict of interest.

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
