# Peer review of "The Influence of Replacing Aggregates and Cement by LFS on the Corrosion of Steel Reinforcements"

_applsci, doi:10.3390/app9040683_

Reviewer 1 Report

It is a good paper, on a topic which is becoming ever more relevant.

The introduction and the bibliography on the state of the art must be completely revised:

1) line 35: ref. 19 refers to blast furnace slag, not to steel slag.

2) line 42: the volumes reported are wrong. From the EUROSLAG site https://www.euroslag.com/products/statistics/statistics-2016/

the volumes produced in 2016 are 

5.9 Mio t Electric Arc Furnace Slag (both carbon steel and stainless steel)
10.4 Mio t Basic Oxygen Furnace Slag
2.0 Mio t Ladle Slag

3) line 44: ref. 25 is from Taiwan. It is not European

4) line 46: ref. 26 is about Electric Arc Furnace slag, ref.s 28 and 30 are about Blast Furnace slag. Moreover Electric Arc Furnace slag is usually referred to as black slag, and ladle slag as white slag. Black ladle slag is an awkward definition. What do the authors mean?

In addition, the authors should consider the following comments: 

5) lines 61-62: the English phrasing of this sentence shall be improved.

6) Table 1 shall be completely redrawn.

7) line 79: which retardant has been used?

8) line 167: chloride ions, not ions chloride.

9) line 182: than, not that.

10) line 259: and, not y

11) line 261:the English phrasing of this sentence shall be improved. 

Author Response

- The introduction and the bibliography on the state of the art must be completely revised:

1) line 35: ref. 19 refers to blast furnace slag, not to steel slag. Horno eléctrico de arco

2) line 42: the volumes reported are wrong. From the EUROSLAG site https://www.euroslag.com/products/statistics/statistics-2016/

the volumes produced in 2016 are

5.9 Mio t Electric Arc Furnace Slag (both carbon steel and stainless steel)
10.4 Mio t Basic Oxygen Furnace Slag
2.0 Mio t Ladle Slag

3) line 44: ref. 25 is from Taiwan. It is not European

4) line 46: ref. 26 is about Electric Arc Furnace slag, ref.s 28 and 30 are about Blast Furnace slag. Moreover Electric Arc Furnace slag is usually referred to as black slag, and ladle slag as white slag. Black ladle slag is an awkward definition. What do the authors mean?

The introduction and the bibliography has been completely revised.

- In addition, the authors should consider the following comments:

5) lines 61-62: the English phrasing of this sentence shall be improved. Rewritten again.

6) Table 1 shall be completely redrawn. Table 1 has been redesigned, separating the physical and chemical characteristics of cement, sand and slag into 3 tables.

7) line 79: which retardant has been used? SikaTard. It has been introduced in the text.

8) line 167: chloride ions, not ions chloride. It has changed.

9) line 182: than, not that. It has changed.

10) line 259: and, not y. . It has changed

11) line 261:the English phrasing of this sentence shall be improved. Rewritten again.

Reviewer 2 Report

The paper deals with the influence of replacing aggregates and cement by LFS on the corrosion of steel reinforcements. The information presented in the paper is valuable. The following minor revisions are recommended.

1. Table 1 is difficult to read.  It should be rearranged and the data should be aligned.

2. Page 2, line 67-72 and page 3, line 76-79 should be rearranged for easier reading. Furthermore, the retardant is 0.5% in the text, but it is shown as 0.5g and 0.35g in the Table 2. Please describe more clearly.

3. There are too many lines in Fig. 3 and Fig. 4, therefore, it affects the reading in different partitions (active state, uncertainty state, and passive state).

4.  Please confirm if there are any typos in the statement in Fig. 8 “…(a) y with LFS (b), with 0.2% chloride…”.

5.  Is the “y” should be “&” in Fig. 9?

Author Response

1. Table 1 is difficult to read. It should be rearranged and the data should be aligned.

Table 1 has been redesigned, separating the physical and chemical characteristics of cement, sand and slag into 3 tables.

2. Page 2, line 67-72 and page 3, line 76-79 should be rearranged for easier reading.

They have been rewritten.

Furthermore, the retardant is 0.5% in the text, but it is shown as 0.5g and 0.35g in the Table 2. Please describe more clearly.

The percentage of retardant in cement weight, which corresponds to 0.5%, is indicated in the text. In the table, the quantities that have been used for each material are indicated, that is why 0.5g or 0.35g appear. The paragraph has been rewritten and has been added "weight of cement" in the text to facilitate its compression.

3.There are too many lines in Fig. 3 and Fig. 4, therefore, it affects the reading in different partitions (active state, uncertainty state, and passive state).

Figures 3 and 4 have been simplified with respect to horizontal lines so that it is better understood.

4. Please confirm if there are any typos in the statement in Fig. 8 “…(a) y with LFS (b), with 0.2% chloride…”.

There was a typographical error that has already been corrected.

5. Is the “y” should be “&” in Fig. 9?

Yes, has already been corrected.
